# An Early Mediterranean-Based Nutritional Intervention during Pregnancy Reduces Metabolic Syndrome and Glucose Dysregulation Rates at 3 Years Postpartum

**DOI:** 10.3390/nu15143252

**Published:** 2023-07-22

**Authors:** Verónica Melero, Maria Arnoriaga, Ana Barabash, Johanna Valerio, Laura del Valle, Rocio Martin O’Connor, Maria Paz de Miguel, Jose Angel Diaz, Cristina Familiar, Inmaculada Moraga, Alejandra Duran, Martín Cuesta, María José Torrejon, Mercedes Martinez-Novillo, Maria Moreno, Gisela Romera, Isabelle Runkle, Mario Pazos, Miguel A. Rubio, Pilar Matia-Martín, Alfonso Luis Calle-Pascual

**Affiliations:** 1Endocrinology and Nutrition Department, Instituto de Investigación Sanitaria del Hospital Clínico San Carlos (IdISSC), Hospital Clínico Universitario San Carlos, 28040 Madrid, Spain; veronica.meleroalvarez10@gmail.com (V.M.); maria.arnoriaga.rodriguez@gmail.com (M.A.); ana.barabash@gmail.com (A.B.); valeriojohanna@gmail.com (J.V.); lauradel_valle@hotmail.com (L.d.V.); rocio@oconnor.es (R.M.O.); pazdemiguelnovoa@gmail.com (M.P.d.M.); joseangeldiazperez561@gmail.com (J.A.D.); cristinafamiliarcasado@gmail.com (C.F.); aduranrh@gmail.com (A.D.); cuestamartintutor@gmail.com (M.C.); irunkledelavega@gmail.com (I.R.); mario_pazos_guerra@hotmail.com (M.P.); marubioh@gmail.com (M.A.R.); 2Facultad de Medicina, Medicina II Department, Universidad Complutense de Madrid, 28040 Madrid, Spain; 3Centro de Investigación Biomédica en Red de Diabetes y Enfermedades Metabólicas Asociadas (CIBERDEM), 28029 Madrid, Spain; 4Clinical Laboratory Department, Instituto de Investigación Sanitaria del Hospital Clínico San Carlos (IdISSC), Hospital Clínico Universitario San Carlos, 28040 Madrid, Spain; mjosetorrejon@gmail.com (M.J.T.); mercedes.martineznovillo@salud.madrid.org (M.M.-N.); maria.moreno.munoz@salud.madrid.org (M.M.); gisela.romera@salud.madrid.org (G.R.)

**Keywords:** abnormal glucose regulation, gestational diabetes mellitus, nutritional intervention, Mediterranean diet, postpartum, cardiovascular disease, metabolic syndrome

## Abstract

A Mediterranean diet (MedDiet)-based intervention reduces the rate of immediate postpartum maternal metabolic disorders. Whether these effects persist long-term remains to be determined. A total of 2526 normoglycemic women were randomized before the 12th gestational week (GW). IG women followed a MedDiet with extra virgin olive oil (EVOO) (>40 mL/day) and a handful of nuts daily, whereas CG women had to restrict all kinds of dietary fat. At 3 months postpartum, a motivational lifestyle interview was held. The endpoint of the study evaluated the rate of abnormal glucose regulation (AGR) and metabolic syndrome (MetS) at 3 years postpartum in women of the San Carlos cohort. A total of 369/625 (59%) CG women and 1031/1603 (64.3%) IG women were finally analyzed. At 3 months and 3 years postdelivery, the IG women showed higher adherence to the MedDiet, which was associated with lower values of body mass index (BMI) and lipid and glycemic profiles. Body weight change and waist circumference were lower in the IG women. After applying multiple regression analysis, the ORs (95%CI) resulted in AGR (3.18 (2.48–4.08); *p* < 0.001)/MetS (3.79 (1.81–7.95); *p* = 0.001) for women with GDM and higher OR for development of MetS in CG women (3.73 (1.77–7.87); *p* = 0.001). A MedDiet-based intervention early in pregnancy demonstrated persistent beneficial effects on AGR and MetS rates at 3 years postpartum.

## 1. Introduction

Gestational diabetes mellitus (GDM) increases the risk of occurrence of type 2 diabetes mellitus (T2DM) later in life [1,2]. Women with a prior history of GDM have an almost 10-times higher risk of developing T2DM than those with normal glucose tolerance (NGT) during pregnancy, especially within the first five years postpartum [3,4]. Nevertheless, different rates of T2DM are found, depending on ethnicity and the diagnostic criteria employed [5]. The importance of a diagnosis of T2DM after delivery lies in its relationship with an increased risk of cardiovascular disease (CVD) [6,7].

The pathogenesis of GDM is complex and multifactorial [8,9,10,11]. However, different factors have been found to be related to its increasing incidence. Obesity, advanced maternal age, smoking habit, and a prior history of GDM seem to contribute to its risk, as well as that of subsequent abnormal glucose regulation (AGR) [8,12,13,14,15]. Ethnicity seems to be another risk factor in the case of certain ethnic groups, as Hispanics are more prone to the development of GDM [16]. Likewise, GDM can increase the predisposition for metabolic syndrome (MetS) [17]. Development of the latter, in turn, increases the risk of T2DM. In addition, other components of MetS, such as lipid disorders and hypertension, are found more frequently in women with a history of GDM [18]. Furthermore, the presence of dyslipidemia in women with a prior diagnosis of GDM is associated with an increased risk of future development of T2DM [19].

Healthy lifestyle and nutritional habits have been found to be factors that can lead to a decrease in the rate of occurrence of GDM. The Mediterranean diet (MedDiet) is one of the most widely studied dietary patterns [20,21]. Prior research conducted by our group on this same cohort of women (the San Carlos cohort) evaluated participants with NGT or GDM after an intervention with MedDiet early in pregnancy and during the immediate postpartum period. The results revealed a 26% decrease in the risk of developing MetS, being more marked in normoglycemic women of the IG compared with those of the control standard-care group [22]. In another study, women with a prior history of GDM were evaluated during pregnancy and within the following 3 years in the context of a lifestyle program. Results indicated a decrease in the risk of developing T2DM [23]. Other studies have also evaluated the relationship between diet and the development of T2DM later in life in women with a prior history of GDM [24]. In fact, several studies have detected that a nutritional intervention early in pregnancy can reduce the risk of GDM, especially in high-risk women [25,26,27].

Thus, the promotion of healthy nutritional habits during pregnancy seems to be decisive, as an early nutritional intervention can decrease health complications in the mother [28]. However, whether these benefits can be sustained has yet to be elucidated. 

The aim of the current study is the assessment of the impact of a nutritional intervention with MedDiet early in pregnancy, reinforced during the postpartum period on the rates of AGR and MetS at 3 years postpartum in women from the San Carlos cohort.

## 2. Materials and Methods

### 2.1. Study Design

This is a prospective unicentric interventional analysis of the “San Carlos GDM prevention study”. This study began in 2015 and its follow-up is ongoing. The study has been undertaken at the Hospital Clínico San Carlos, providing medical assistance to 463.833 patients in Madrid, Spain. During pregnancy, women are followed up from the 8th to 12th gestational week (GW) when the first ultrasound is performed. Between the 24th and 28th GW, the 75 g oral glucose tolerance test (OGTT) is carried out for detection of GDM. Both the nursing staff and the obstetricians give medical recommendations and nutritional advice to pregnant women from the onset of pregnancy. Since 2012, IADPSG criteria have been applied for the diagnosis of GDM [29].

The first study (Study 1), whose participants have been included in the current one, was a randomized controlled trial (RCT), with women allocated before the 12th GW into the control group (CG) and the intervention group (IG). A MedDiet pattern was recommended: women of the IG were urged to increase the consumption of extra virgin olive oil (EVOO) >40 mL/day and to consume a handful of pistachios daily, with both foods provided at no cost. Conversely, CG women were encouraged to consume less than 40 mL/day of EVOO and to not eat nuts more than three times a week. The second study (Study 2) consisted in evaluating the effects of the MedDiet implemented early in pregnancy on a single group based on routine clinical practice (real world). Women from Study 2 were encouraged to follow the same nutritional indications as the IG from the RCT of Study 1 but were not provided EVOO and pistachios at no cost and were free to choose the nuts they preferred. The third study (Study 3) was an RCT in which women with a body mass index (BMI) ≥ 25 kg/m^2^ were randomized into CG and IG, the latter receiving a recommendation of increasing the consumption of pistachios and EVOO. However, the pistachios were not provided free-of-cost. The IG from both RCTs, as well as the group belonging to the study based on real practice, were analyzed as IG for this study, since these groups were urged to follow a MedDiet pattern along with a specific dietary recommendation in relation to nuts and EVOO. They were compared with the 2 CGs belonging to the RCTs. Women diagnosed with GDM were followed by the Endocrinology Department and received identical treatment per department protocols, regardless of whether they had been allocated to the IG or CG. This information is displayed in Figure 1.

After randomization of participants at the 8th–10th GW, three visits during gestation were undertaken. The first visit was made between the 12th and 14th GW, the second between the 20th and 24th, and the third between the 36th and 38th GW. After delivery, a 3-month visit was made, in which a 60 min motivational interview was carried out to encourage the subjects to continue with the nutritional and lifestyle habits acquired during pregnancy. Detailed information on each visit during and after childbirth is displayed in Figure 2.

All women belonging to previous studies were invited to attend a follow-up visit 3 years postpartum. Participants made an appointment by phone as well as by email/letter. They were informed of the timing of the follow-up as well as the volunteer character of the analysis. Participants who could not attend were invited to complete questionnaires via email or telephone. However, within this 3-year time period, several women were unable to participate in the monitoring for different reasons (such as having a new gestation during the follow-up, or any health condition or treatment that could result in modifications of analytic values and/or in body composition).

### 2.2. Study Population

All the participants analyzed in this study belong to the cohort of the “San Carlos GDM prevention study”. The trials they were included in were previously registered as ISRCTN84389045, ISRCTN13389832, and ISRCTN16896947, were approved by the Ethics Committee of Hospital Clínico San Carlos (ethic codes CI 13/296-E, CI 16/442-E, and CI 16/316), and conducted according to the Helsinki Declaration. All women signed a letter of informed consent. The three studies were carried out consecutively. A total of 2526 normoglycemic women (fasting serum glucose (FSG) < 92 mg/dL) were allocated between the 8th and 12th GW into 2 groups. A total of 2228 women were evaluated at the end of pregnancy (625 women in the CG and 1603 in the IG). Participants were followed up during gestation and postpartum from 2015 to 2018. Throughout the years 2017–2020, women belonging to all groups were followed up at a 3-year postpartum visit. Hence, a total of 369/625 (59%) CG and 1031/1603(64.3%) IG women completed all the visits and were finally included in the analysis at both 3 months and 3 years postpartum. Table 1 shows baseline characteristics.

### 2.3. Sociodemographic, Anthropometric, and Clinical Data

Information on age, ethnicity (Caucasian/Hispanics and others), family history of T2DM and/or MetS, history of previous GDM and/or miscarriages, educational status (holding a university degree), being primiparous, and employment status was collected at the first visit. Pre-pregnancy body weight (BW) and pre-pregnancy BMI was also registered at the first visit. Fat mass (FM) (kg), BMI (kg/m^2^), and BW (kg), measured without shoes and with lightweight clothes, were provided by electrical bioimpedance analysis (SECA mBCA 514) and registered at 3 years postpartum. Waist circumference (WC) was taken using a non-extendable measuring tape. It was measured according to ISAK guidelines [30,31] between the last rib and the iliac crest, slightly above the navel. Blood pressure (BP) was measured using a digital sphygmomanometer with an adequate armlet after a rest period of 10 min in a sitting position (Omron 705IT, Omron Global, Kyoto, Japan). Current diagnoses and medication were registered at the first visit, as well as at the 3-month and 3-year postpartum visits, as was smoking habit (never or current smoker).

### 2.4. Biochemical Analysis

Blood and urine samples were collected early in the morning after a minimum 8 h fast.

The following data were determined: Total cholesterol (T-Chol) was quantitatively determined by the colorimetric cholesterol enzymatic test method (CHOD-PAP). Serum levels of HDL-cholesterol were measured by the enzymatic immunoinhibiting method in an Olympus 5800 (Beckman-Coulter, Brea, CA, USA). LDL-cholesterol was calculated with the Friedewald formula. Serum triglycerides were measured with a colorimetric enzymatic method using glycerol phosphate oxidase p-amino phenazone (GPO-PAP). Dimension Vista (Siemens Healthcare Diagnostics, Munich, Germany) was used to measure apolipoprotein B and C-RP by immunonephelometry and nephelometry, respectively. Fasting serum insulin (FSI) was measured by a chemiluminescence immunoassay in an IMMULITE 2000 Xpi (Siemens, Healthcare Diagnostics, Munich, Germany), with an inter-assay accuracy in concentrations of 11 uIU/mL of 6.3% and for insulin concentration of 21 uIU/mL of 5.91. HOMA-IR was calculated as glucose (mmol/L) × insulin (µUI/mL)/22.7. FSG (glucose oxidase) and HbA1c (%) levels were standardized by the International Federation of Clinical Chemistry and Laboratory Medicine, using ion-exchange high-performance liquid chromatography in gradient, with a Tosoh G8 analyzer (Tosoh Co., Tokyo, Japan). Inter-assay imprecision of HbA1c for levels of 5.1% had standard deviation (SD) of 0.06 and coefficient of variation (CV) of 1.23%; for levels of HbA1c 10.39%, SD was 0.11 and CV was 1.04. An external quality guarantee program of the SEQC (Sociedad Española de Química Clínica) evaluated the quality of the methods monthly.

### 2.5. Dietary Assessment

Lifestyle and MedDiet adherences were evaluated by means of two semiquantitative validated questionnaires. The Mediterranean diet adherence screener (MEDAS) was used to assess adherence to MedDiet. This questionnaire, derived from the PREDIMED study [32], was composed of 14 points. However, only 12 points were applied, as alcohol and juices were excluded from the diet during pregnancy. The compliance with each item was provided as +1. Although a score >5 indicated adequate compliance with the MedDiet, a score >7 was preferable. The second questionnaire applied was the diabetes nutrition and complications trial (DNCT). It collected information on the consumption of specific food groups and physical activity habits. The DNCT questionnaire was made up of 15 items; the first 3 collected information on physical activity and the remaining 12 pertained to the frequency of intake of specified foods. The answers from participants were valued as follows: A (value +1), B (value 0), and C (value −1). Obtention of an A was considered to be a preventive factor for T2DM, whereas obtaining a C reflected an increased risk. Additional information on the MEDAS and DNCT questionnaires used is available in previous publications [25,33].

### 2.6. Outcome Analysis

The endpoint of this study was to evaluate the rates of AGR and MetS at 3 years postpartum.

AGR was defined as impaired fasting serum glucose (IFG) ≥ 100 (mg/dL) and/or impaired glucose tolerance (IGT) by 2 h serum glucose during 75-g OGTT ≥ 140 (mg/dL) and/or impaired HbA1C when rates were ≥5.7%, according to the ADA criteria [34]. Insulin resistance (IR) was diagnosed when the homeostasis assessment model for insulin resistance (HOMA-IR) was ≥3.5, according to cut-off values described in the Spanish population [35,36].

MetS diagnosis and each of its components were evaluated by applying the international harmonized criteria for the diagnosis of MetS [37]. MetS diagnosis was considered when patients showed three or more of the following: a waist circumference (WC) (cm) ≥ 89.5, an FSG (mg/dL) ≥ 100 and/or a HbA1C (%) ≥ 5.7, systolic blood pressure (sBP) (mmHg) ≥ 130 mmHg and/or diastolic blood pressure (dBP) (mmHg) ≥ 85 mmHg, HDL-cholesterol (mg/dL) < 50, and triglycerides (g/L) ≥ 150.

### 2.7. Statistical Analysis

Data were presented as the median and interquartile range (IQR) or numbers (%). Comparisons between groups for categorical variables were evaluated using the chi 2 test; the Mann–Whitney-U test was applied for continuous variables expressed as median (IQR). The magnitude of the association between study groups (CG vs. IG, using CG as the reference group, or GDM group vs. NGT group, using NGT as the reference group) and binary outcomes (IR, AGR, and each MetS component) was evaluated using the relative risk (RR) and 95% confidence interval (CI).

Multivariate logistic regression analyses were conducted to estimate the odds ratios (ORs) and 95% confidence intervals (95% CIs) for AGR and MetS at 3 years postpartum. They were adjusted for maternal age and pregestational BMI. Continuous variables were categorized as ≥35, <35 and >30 kg/m^2^, and <30 kg/m^2^, respectively. ORs were estimated for GDM, CG, and Hispanic ethnicity, pregestational obesity (BMI > 30 kg/m^2^), and 3-years BW > pregestational BW. All analyses were carried out using SPSS, version 21 (SPSS, IBM, Chicago, IL, USA).

## 3. Results

Table 2 displays the biochemical, anthropometric, and clinical data of the women belonging to the IG (*n* = 1031) and CG (*n* = 369) at 3 months and 3 years postpartum.

At 3 months postpartum, adherence to the MedDiet given by the MEDAS score was higher in the IG. A higher MEDAS score was associated with significantly lower BMIs, total cholesterol, LDL-cholesterol, APO-B levels, and CPR. In relation to the glycemic profile, FSI and FSG levels, as well as HOMA-IR, were also lower in the IG. At 3 years postpartum, women in the IG presented a significant reduction in the same parameters as at 3 months postpartum, with the exception of FSG. Additionally, they showed significantly lower values of changes in BW in relation to pregestational values, WC, dBP, HDL-cholesterol, 2H-OGTT, and physical activity score.

Table 3 shows the RR for developing MetS and each component of the MetS in relation to the nutritional intervention.

When analyzing each component of the MetS at 3 months postpartum, no significant differences were found, except for IFG, where belonging to the CG showed a higher risk. When evaluating at 3 years postpartum, the RR of developing prediabetes was higher for women in the CG, who also showed an increased risk of elevated WC, dBP, triglycerides, decreased HDL-cholesterol, and having >2 MetS components.

The RR of developing MetS and each of its components was assessed in women with a diagnosis of GDM. At 3 months postpartum, the risk of developing each MetS component was higher in women with GDM, with the exception of raised triglyceride levels, decreased HDL-cholesterol levels, and elevated HOMA-IR, where no significant differences were observed. At 3 years postpartum, each MetS component in women with a diagnosis of GDM was significantly higher, except for triglycerides, HDL-cholesterol levels, and dBP. The risk of developing >2 components of MetS was higher in women with a diagnosis of GDM in both postnatal periods. 

After applying multivariate logistic regression, the OR (95% CI) to present AGR and/or MetS, respectively, was: (1.82 (1.30–2.54), *p* = 0.000 and 2.75 (1.22–6.18), *p* = 0.04) for Hispanic ethnicity, (1.99 (1.19–3.32), *p* = 0.009 and 4.06 (1.58–10.40) *p*= 0.009) for women with pre-pregnancy obesity, and (0.99 (0.98–1.01) *p*= 0.284 and 1.50 (1.09–2.06), *p* = 0.007) for women >35 years of age at the 12th GW. The OR for developing AGR and/or MetS in women with a diagnosis of GDM was (3.18 (2.48–4.08), *p* < 0.001) and (3.79 (1.81–7.95), *p* = 0.001), respectively. The statistically significant OR in women belonging to the CG was solely in relation to the development of MetS (3.73 (1.77–7.87), *p* = 0.001). This information can be found in Figure 3.

## 4. Discussion

A MedDiet-based nutritional intervention enriched with EVOO and nuts implemented early in pregnancy (before the 12th GW) and maintained throughout the postpartum period in the IG women was accompanied by a higher MEDAS score, which was in turn associated with lower rates of AGR and MetS at 3 years postdelivery compared with the CG women who initiated the intervention after delivery. Women in the IG presented more risk factors for the development of AGR and MetS, such as older age and a more frequent history of gestational adverse events (miscarriages and/or diagnosis of GDM). Despite these factors, the rates of AGR and/or MetS were lower. This fact reinforces the importance of starting the nutritional intervention as early as possible during pregnancy to reduce the possibility of an unfavorable postnatal metabolic impact. The response rate to postnatal follow-up was about 60% of the participants, as expected, indicating the difficulty in carrying out postnatal follow-up as recommended by all scientific societies.

A recent systematic review also highlighted that lifestyle and nutritional interventions maintained in the postpartum period can induce changes in glucose regulation [38]. Especially relevant is when diet is started early in pregnancy, focusing on women at high risk [26,27]. Several trials have addressed the association between diet started in pregnancy and maintained postdelivery and the development of T2DM in women with a diagnosis of GDM [39,40,41]. Other studies have also assessed the influence of diet in the postpartum period on women with pre-pregnancy obesity and a previous history of GDM, obtaining successful outcomes [24,42]. Our results coincide with the aforementioned studies regarding the impact of diet in the reduction of the risk in developing postpartum prediabetes, especially regarding high-risk women. However, to our knowledge, this is the first study that has evaluated a MedDiet-based nutritional intervention early in pregnancy in a population study (both NGT and GDM women) over a long follow-up period in relation to glucose dysregulation at 3 years postpartum. In addition to showing greater adherence to the MedDiet, the women who received the nutritional intervention from the beginning of pregnancy compared with those who received it after childbirth had a lower BMI and waist circumference, gained less body weight in relation to the pre-pregnancy body weight, and had a lower value of HOMA-IR. These data suggest that adherence to the MedDiet is associated with an improvement in IR. In this regard, one of the factors involved in the progression towards MetS and/or AGR is IR [43]. The risk of developing these diseases can increase further if pancreatic insulin secretion is reduced as a consequence of impairment in beta-cell function [44]. In the current study, women presented a lower value in the 2 h-OGTT glucose levels, indicating the preservation of the beta cell and its secretory capacity.

Considering the specific cut-off points to determine each of the MetS and/or AGR components, GDM emerged as the main risk factor associated with MetS and AGR development at 3 years postpartum. Although the rates of MetS and AGR were substantially reduced when comparing prior studies [22,23,45] to the current analysis, they remained high. Furthermore, the risk of later development of T2DM seemed to increase within the first years after delivery [46]. The results of the current study indicate that in women with a diagnosis of GDM, the risk of having each component of MetS and an overall risk of developing MetS is higher than in women with NGT. Furthermore, an association is observed between a smaller weight gain and lower levels of HOMA in IG women. This could explain lower rates of AGR at 3 years postdelivery in the IG due to greater adherence to the MedDiet patterns both during pregnancy and after delivery. This reduction in HOMA levels could in turn explain the decrease in the risk of developing MetS and its components. The role of the MedDiet in the prevention of MetS and AGR may be due to its antioxidant characteristics and other anti-inflammatory aspects [47]. EVOO is a rich source of monounsaturated fatty acids and has been found to lower postprandial glucose levels as well as to improve the inflammatory profile; it could also limit weight gain by reducing the carbohydrate load of meals. Nut consumption may facilitate weight loss within energy-restricted diets, possibly due to enhanced satiety, increased thermogenesis, incomplete mastication, and fat malabsorption Nuts are rich in unsaturated fatty acids, fiber, magnesium, and other phytochemical constituents, with potential beneficial effects on insulin sensitivity, fasting glucose levels, and inflammation. Therefore, a reduction in the MetS and AGR rates could be expected with the MedDiet.

Other established risk factors, such as Hispanic ethnicity, maternal age, pregestational obesity, and a prior history of GDM or family history of T2DM, were also considered in the current study. In this analysis, only the Hispanic race was found to be associated with a higher rate of MetS and AGR at 3 years postpartum. Previous studies have shown a higher risk of the development of GDM in Hispanic women compared with Caucasian women, as well as postpartum AGR or MetS. Similarly, a family history of diabetes mellitus has also been found to increase the risk of postpartum prediabetes in women with a diagnosis of GDM [16,48,49]. In the current study, pre-pregnancy obesity rates (≥30 kg/m^2^) were probably insufficient to achieve statistical significance, but when we adjusted OR for pre-pregnancy BMI and age, as continuous variables, only GDM was associated with more than a 3-fold increased risk of developing AGR and/or MetS.

This study has certain limitations. First, the differences in relation to the MEDAS score between the IG and CG were small, since MedDiet-based nutritional recommendations were provided to both groups (similarly, supplementation or restriction in the consumption of nuts and EVOO). Thus, small or absent differences in the questionnaire scores were to be expected. Secondly, the IG was larger than the CG. Moreover, IG women were older and had more previous adverse events; thus, worse outcomes could be expected in the IG, whereas the results indicated the opposite, further supporting the benefits of the nutritional intervention. Lastly, semiquantitative questionnaires may not always be accurate in terms of the responses obtained. However, they are validated, are the instruments most frequently used to obtain a vision of eating habits, and are always applied by specialized professionals to minimize bias. The greatest strength of our study lies in the study of both populations, NGT women and those with GDM, since most studies only evaluate nutritional intervention in women who have developed GDM.

## 5. Conclusions

In summary, our study shows that a nutritional intervention based on the principles of the MedDiet and supplemented with EVOO and nuts initiated early in pregnancy not only reduces the rate of GDM but also has metabolic benefits up to 3 years postpartum when compared with a nutritional intervention commenced after delivery. Therefore, easily adopted eating patterns, such as an increase in the consumption of EVOO and pistachios, maintained over time, improve clinical parameters and can act as a preventive factor for later development of T2DM. They can help to reduce the risk of GDM and in turn a later progression to T2DM. Postnatal MedDiet-based nutritional interventions should be intensified in women with GDM. Further studies will be needed to evaluate whether these benefits persist for more than three years.

## Figures and Tables

**Figure 1 nutrients-15-03252-f001:**
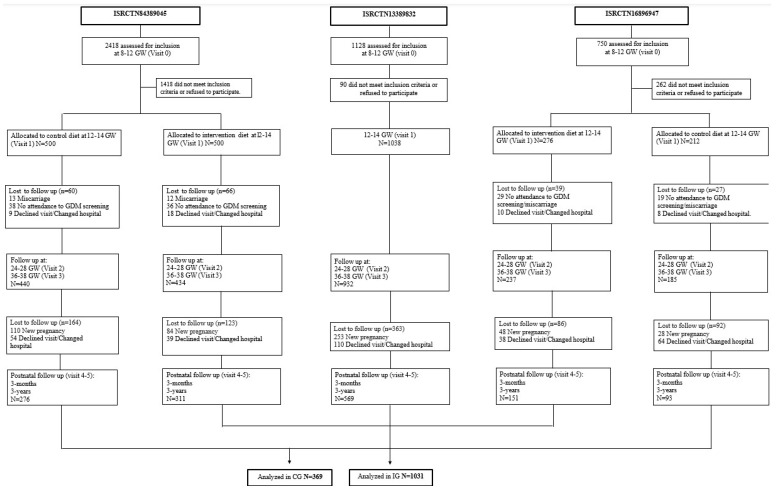
Flowchart of the cohorts included in the analyses. Abbreviations: GDM—gestational diabetes mellitus; GW—gestational weeks.

**Figure 2 nutrients-15-03252-f002:**
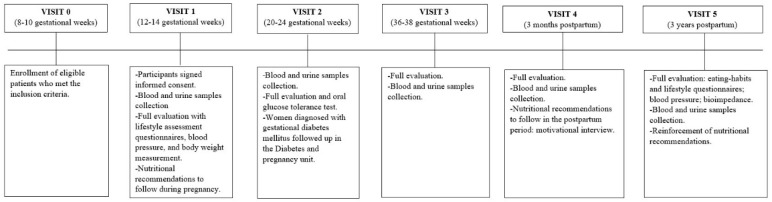
Description of the intervention performed at each visit during pregnancy and postpartum.

**Figure 3 nutrients-15-03252-f003:**
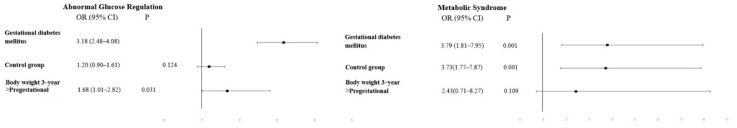
Multiple logistic regression adjusted by maternal age and body mass index at 3 years postdelivery. Abbreviations: OR—odds ratios; CI—confidence intervals; AGR—abnormal glucose regulation (*p* > 0.05).

**Table 1 nutrients-15-03252-t001:** Baseline characteristics of analyzed women.

	CG *N* = 369	IG *N* = 1031	*p*
Age (Years)	33 (29–36)	34 (30–37)	0.032
Race/Ethnicity			
Caucasian	238 (64.8%)	697 (67.7%)	0.317
Hispanic	117 (31.9%)	313 (30.4%)	
Others	14 (3.3%)	21 (1.9%)	
Family History of			
Type 2 Diabetes	13 (3.5%)	41 (4.0%)	0.333
Mets (>2 Components)	66 (17.9%)	215 (20.9%)	
Previous History of			
Gdm/Miscarriages	12 (3.3%)/112 (30.4%)	37 (3.7%)/346 (33.6%)	0.038
Educational Status			
University Degree	232 (62.9%)	734 (71.3%)	0.005
Employment	292 (79.6%)	842 (81.7%)	0.035
Number of Pregnancies			
Primiparous	161 (43.6%)	464 (45.2%)	0.365
Smoker			
Never	201 (54.6%)	600 (58.2%)	0.359
Current	26 (7.1%)	69 (6.7%)	
Pre-Pregnancy Body Weight (kg)	60 (54–68)	60 (54–67)	0.173
Pre-Pregnancy BMI (kg/m^2^)	23.1 (20.6–26.0)	23.0 (20.5–25.9)	0.164

**Table 2 nutrients-15-03252-t002:** Postnatal biochemical, anthropometric, and clinical data at 3 months and 3 years postdelivery by groups.

	CG (369)		IG (1031)		*p* (CG vs. IG)	
	3 Months PD	3-Year PD	3 Months PD	3-Year PD	3-M	3 Years
BW (kg)	66.5 (58.7–75.9)	63.3 (56.0–70.8)	64.9 (58.0–72.0)	62.0 (55.6–68.9)	0.108	0.604
BMI (kg/m^2^)	25.3 (22.3–28.7)	24.0 (21.3–27.1)	24.2 (21.9–27.1)	23.2 (21.1–25.7)	0.015	0.025
BW-Change (kg)	4.5 (1.9–7.4)	2.5 (−0.1–5.9)	4.4 (1.6–7.5)	2.0 (−5.2–3.3)	0.707	0.045
WC (cm)	85 (78–93)	80 (72–89)	85 (79–92)	79 (73–87)	0.754	0.009
Fat mass (kg)	Na	20.1 (14.3–25.2)	Na	20.1 (15.6–24.8)	—	0.676
sBP (mmHg)	111 (103–118)	111 (104–118)	110 (102–119)	111 (102–119)	0.826	0.549
dBP (mmHg)	71 (65–76)	74 (68–79)	70 (65–76)	70 (66–75)	0.927	0.003
T-Chol (mg/dL)	200 (176–224)	179 (158–192)	192 (174–217)	171 (156–196)	0.006	0.048
HDL-CHOL (mg/dL)	62 (55–71)	55(49–63)	61 (54–72)	58 (51–66)	0.881	0.004
LDL-Chol (mg/dL)	122 (103–144)	112 (87–117)	113 (96–133)	102 (86–116)	0.001	0.002
Triglycerides (g/L)	73 (55–97)	72 (52–98)	68 (54–92)	67 (54–87)	0.050	0.058
Apo B (mg/dL)	89 (75–103)	79 (69–92)	83 (73–97)	79 (69–91)	0.007	0.681
FSI (μIU/mL)	6.1 (3.9–10.3)	7.4 (4.3–10.9)	4.5 (2.6–8.0)	5.4 (3.1–8.9)	0.000	0.009
HOMA-IR	1.4 (0.9–2.2)	1.8 (1.0–2.8)	1.1 (0.7–2.0)	1.4 (1.0–2.3)	0.001	0.008
FS Glucose (mg/dL)	84 (80–90)	90 (85–96)	82 (79–89)	89 (83–95)	0.040	0.195
2h-OGTT (mg/dL)	Na	98 (85–119)	Na	94 (81–109)	—	0.037
HbA1c-IFCC %	5.3 (5.1–5.5)	5.3 (5.1–5.5)	5.3 (5.1–5.5)	5.4 (5.2–5.5)	0.635	0.132
cPR (mg/dL)	0.20 (0.10–0.44)	0.14 (0.04–0.29)	0.15 (0.07–0.31)	0.11 (0.05–0.29)	0.005	0.020
Phisycal activity Score	−2 (−2;−1)	−2 (−2;−1)	−2 (−2;−1)	−1 (−2;−1)	0.801	0.006
Nutrition Score	4 (1; 6)	2 (−1;4)	4 (2; 7)	1 (0;5)	0.200	0.013
MEDAS Score	6 (5–7)	7 (5–8)	6 (5–8)	7 (6–8)	0.002	0.047

Data are median (IQR). Body weight (BW); body mass index (BMI); waist circumference (WC); not available (Na); systolic blood pressure (sBP); diastolic blood pressure (dBP); total cholesterol (T-chol.); high-density lipoprotein (HDL); low-density lipoprotein (LDL) fasting serum insulin (FSI); lipoprotein B (APO-B); fasting serum glucose (FS glucose); homeostasis assessment model for insulin resistance (HOMA-IR); C reactive protein (CPR); Mediterranean diet adherence screener (MEDAS). Physical activity score, (walking daily (>5 days/week): score 0: at least 30 min; score +1, if >60 min; score −1, if <30 min. Climbing stairs (floors/day, >5 days a week): score 0, between 4 and 16; score +1, >16; score −1: <4).

**Table 3 nutrients-15-03252-t003:** Comparison of postdelivery rate of metabolic syndrome (MetS) components between women from the intervention group (IG) vs. the control group (CG) and women with gestational diabetes mellitus (GDM) vs. normal glucose tolerance (NGT). Panel A, 3 months postpartum. Panel B, 3 years postpartum.

	CG (369) vs. IG (1031)			GDM (290) vs. NGT (1110)		
	*N* (%)	RR (95% CI) IG	*p*	*N* (%)	RR (95% CI) GDM	*p*
**Panel A. 3 months**						
Glycemic Status						
IFG	18 (4.9) vs. 29 (2.8)	0.68 (0.47–0.98)	0.046	19 (6.6) vs. 28 (2.6)	1.33 (1.05–1.69)	0.002
Prediabetes (HBA1c ≥ 5.7%)	20 (5.4) vs. 71 (6.9)	1.29 (0.72–2.33)	0.242	30 (10.1) vs. 61 (5.3)	1.25 (1.03–1.52)	0.006
MetS components						
Raised (WC ≥ 89.5 cm)	69 (18.7) vs. 202 (19.3)	1.06 (0.78–1.44)	0.386	99 (34.1) vs.172 (15.5)	1.30 (1.18–1.42)	0.000
Raised sBP ≥ 130 mmHg	13 (3.5) vs. 52 (5.0)	1.46 (0.78–2.70)	0.147	22 (7.6) vs. 43 (3.9)	1.26 (1.00–1.38)	0.033
Raised dBP ≥ 85 mmHg	16 (4.3) vs. 52 (5.0)	1.17 (0.66–2.08)	0.351	25 (8.6) vs. 43 (3.9)	1.22 (1.02–1.46)	0.007
Raised TRIG ≥ 150 mg/dL	26 (7.0) vs. 68 (6.5)	0.93 (0.58–1.49)	0.427	26 (8.7) vs. 68 (6.1)	1.08 (0.95–1.22)	0.125
Reduced HDL-C < 50 mg/dL	35 (9.5) vs. 92 (8.9)	0.94 (0.62–1.41)	0.409	38 (13.1) vs. 89 (8.0)	1.08 (0.96–1.20)	0.098
AGR	30 (8.1) vs. 98 (9.5)	1.05 (0.65–1.69)	0.435	46 (16.2) vs. 82 (7.4)	1.31 (1.10–1.55)	0.000
Raised HOMA-IR ≥ 3.5	21 (5.7) vs. 46 (4.5)	0.77 (0.46–1.32)	0.308	18 (6.2) vs. 49 (4.4)	1.05 (0.91–1.21)	0.303
>2 componets of MetS	22 (5.8) vs. 54 (5.3)	0.94 (0.47–1.90)	0.493	32 (10.6) vs. 44 (4.0)	1.40 (1.07–1.84)	0.002
**Panel B. 3 Years**						
Glycemic Status						
IFG	47 (12.8) vs.102(9.9)	0.86 (0.59–1.27)	0.254	77 (30.3) vs. 72 (6.5)	1.64 (1.39–1.94)	0.000
Prediabetes (HbA1c ≥ 5.7%)	58 (13.7) vs. 19 (1.7)	0.90 (0.80–1.00)	0.018	34 (11.8) vs. 43 (3.9)	1.41 (1.15–1.73)	0.000
IGT	4 (1.2) vs. 9 (0.9)	0.72 (0.21–2.46)	0.403	10 (3.4) vs. 3 (0.3)	2.97 (1.13–7.79)	0.000
MetS components						
BMI ≥ 30 (kg/m^2^)	28 (7.7) vs. 62 (6.1)	0.79 (0.49–1.24)	0.175	40 (13.8) vs. 46 (4.1)	1.73 (1.38–2.17)	0.000
Raised (WC ≥ 89.5 cm)	34 (9.2) vs. 45 (4.4)	0.55 (0.33–0.92)	0.017	29 (10.0) vs. 50 (4.5)	1.23 (1.04–1.47)	0.003
Raised sBP ≥ 130 mmHg	6 (1.8) vs. 21 (2.1)	1.04 (0.40–2.72)	0.574	14 (4.8) vs. 13 (1.2)	1.43 (1.01–2.05)	0.020
Raised dBP ≥ 85 mmHg	35 (9.3) vs. 3 (0.5)	0.76 (0.69–0.84)	0.001	8 (2.8) vs. 30 (2.7)	1.02 (0.86–1.20)	0.480
Raised TRIG ≥150 mg/dL	32 (8.6) vs. 43 (4.2)	0.59 (0.36–0.96)	0.023	26 (9.0) vs. 49 (4.4)	1.19 (0.98–1.26)	0.131
Reduced HDL-C < 50 mg/dL	78 (21.1)vs.153 (14.8)	0.83 (0.67–0.98)	0.048	67 (21.7) vs.164 (14.8)	1.09 (0.99–1.14)	0.103
AGR	8 (22.0) vs. 112 (10.9)	0.97 (0.76–1.24)	0.439	89 (30.1) vs.104 (9.4)	1.52 (1.33–1.73)	0.001
Raised HOMA-IR ≥ 3.5	61 (16.6) vs. 118 (11.5)	0.66 (0.38–1.17)	0.100	56 (19.4) vs.123 (11.1)	1.22 (1.00–1.50)	0.022
>2 componets of MetS	17 (4.5) vs. 15 (1.5)	0.51 (0.36–0.76)	0.003	18 (6.2) vs. 14 (1.3)	1.56 (1.09–2.25)	0.001

Data are median (IQR) or number (%). Impaired fasting glucose (IFG); metabolic syndrome (MetS); high-density lipoprotein (HDL); waist circumference (WC); systolic blood pressure (sBP); diastolic blood pressure (dBP); triglycerides (TRG); abnormal glucose regulation (AGR); impaired glucose tolerance (IGT); body mass index (BMI); GDM: gestational diabetes mellitus, NGT: normal glucose tolerance; control group (CG); intervention group (IG). P denotes differences between groups.

## Data Availability

The data analyzed in this study are subject to the following licenses/restrictions: no restriction. Requests to access these datasets should be directed to acalle.edu@gmail.com.

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
