# Peer review of "An Early Mediterranean-Based Nutritional Intervention during Pregnancy Reduces Metabolic Syndrome and Glucose Dysregulation Rates at 3 Years Postpartum"

_nutrients, 2023, doi:10.3390/nu15143252_

Round 1
Reviewer 1 Report
This is an interesting and well-conducted study aimed to evaluate the impact of a nutritional intervention (Med Diet) since early in pregnancy and reinforced postpartum on the rates of abnormal glucose regulation and metabolic syndrome at 3 years post-partum in women from the San Carlos cohort. Those two conditions are mainly responsible for women's morbidity and mortality as life progresses. Thus, investigating simple strategies to reverse this is always of high health value and research towards this goal should be supported and become broadly projected. Review of the literature is adequate, methods and analysis is valid, results are clearly presented and conclusions are supported by the results. Limitations are also reported though of minor effect.
Author Response
Thank you very much for your kind comments.
It does not suggest making any modifications

Reviewer 2 Report
This manuscript proves that Mediterranean diet (MedDiet)-based nutritional intervention during pregnancy reduces metabolic disorders with over 1400 participants. This manuscript provides new findings about usefulness of MedDiet to reduce metabolic disorders in pregnant women.
There are some questions as follows.
1. Why do you set the amount of extra virgin olive oil (EVOO) supplementation at less than >40ml/day?.
2. BMI in IG decreased compared to that in CG. Do you think MedDiet-based nutritional intervention prevent obesity in pregnant women?
3. The beneficial effects of the MedDiet can be mainly attributed to it numerous components rich in anti-inflammatory and antioxidant properties (Nutrients 2021, 13, 1951. doi.org/10.3390/nu13061951). The mechanism of reducing metabolic syndrome and glucose dysregulation rates in pregnant women by early MedDiet-based nutritional intervention (authors’ speculation) should be mentioned in Discussion.
Author Response
Thank you very much for your kind comments and constructive suggestions. We agree with you point-by-point, and have tried to make the corresponding changes to improve the article. The changes applied are as follows:
This manuscript proves that Mediterranean diet (MedDiet)-based nutritional intervention during pregnancy reduces metabolic disorders with over 1400 participants. This manuscript provides new findings about usefulness of MedDiet to reduce metabolic disorders in pregnant women. There are some questions as follows.
- Why do you set the amount of extra virgin olive oil (EVOO) supplementation at less than >40ml/day?.
This amount is what the MEDAS questionnaire (2nd items) collects for adherence to the Mediterranean diet, and is used in our study and in others (PREDIMED).
|
2. How much olive oil do you consume in a given day (including oil used for frying, salads, out-of-house meals,)? Criteria for 1 point: ≥4 tbsp (40ml/day) |
- BMI in IG decreased compared to that in CG. Do you think MedDiet-based nutritional intervention prevent obesity in pregnant women?
Indeed, the nutritional intervention based on the Mediterranean diet is associated with less weight gain during pregnancy, and we have previously published this observation (Gestational Diabetes Mellitus treatment reduces obesity induced adverse pregnancy and neonatal outcomes. The St Carlos Gestational Study, BMJ Open Diabetes Research and Care doi:10.1136/bmjdrc-2016- 000314). Although there is a higher consumption of EVOO and nuts, this is associated with a higher consumption of vegetables with a low glycemic load and lower caloric density. On the other hand, the consumption of nuts can also be considered. Nuts may facilitate weight loss within energy-restricted diets, possibly due to enhanced satiety, increased thermogenesis, incomplete mastication and fat malabsorption. Nuts are rich in unsaturated fatty acids, fiber, magnesium, and other phytochemical constituents with potential beneficial effects on insulin sensitivity, fasting glucose levels and inflammation.
- The beneficial effects of the MedDiet can be mainly attributed to it numerous components rich in anti-inflammatory and antioxidant properties (Nutrients 2021, 13, 1951. doi.org/10.3390/nu13061951). The mechanism of reducing metabolic syndrome and glucose dysregulation rates in pregnant women by early MedDiet-based nutritional intervention (authors’ speculation) should be mentioned in Discussion.
We agree with your comments. Increased EVOO and nuts consumption are clearly beneficial. EVOO is a rich source of monounsaturated fatty acids, and has been found to lower postprandial glucose levels as well as to improve the inflammatory profile. EVOO could have limited weight gain by reducing the carbohydrate load of meals. Furthermore, its liberal use facilitates an increased intake of vegetables, traditionally eaten with olive oil in Spanish cuisine.
Nuts facilitate weight loss within energy-restricted diets, possibly due to enhanced satiety, increased thermogenesis, incomplete mastication and fat malabsorption. Nuts are rich in unsaturated fatty acids, fiber, magnesium, and other phytochemical constituents with potential beneficial effects on insulin sensitivity, fasting glucose levels and inflammation. Their antioxidant capacity of pistachios is higher than other nuts, given their high levels of lutein,β-carotene, and γ-tocopherol. Pistachio consumption improves the inflammatory cytokine profiles linked to GDM development.
According to your comment we have inserted the following sentence in the discussion and add the reference (47: Gantenbein, K.V.; Kanaka-Gantenbein, C. Mediterranean Diet as an Antioxidant: The Impact on Metabolic Health and Overall Wellbeing. Nutrients 2021, 13, 1951. https://doi.org/10.3390/nu13061951) :The role of the MedDiet in the prevention of MetS and AGR may be due to its antioxidant characteristics and other anti-inflammatory aspects (47). EVOO is a rich source of monounsaturated fatty acids, and has been found to lower postprandial glucose levels as well as to improve the inflammatory profile, and could have limited weight gain by reducing the carbohydrate load of meals. Nuts consumption may
facilitate weight loss within energy-restricted diets, possibly due to enhanced satiety,
increased thermogenesis, incomplete mastication and fat malabsorption Nuts are rich in
unsaturated fatty acids, fiber, magnesium, and other phytochemical constituents with
potential beneficial effects on insulin sensitivity, fasting glucose levels and
inflammation. Therefore a reduction in the MetS and AGR rate could be expected with
the MedDiet

Round 2
Reviewer 2 Report
This manuscript is revised well.